# A machine learning enhanced EMS mutagenesis probability map for efficient identification of causal mutations in *Caenorhabditis elegans*

**Zhengyang Guo, Shimin Wang, Yang Wang, Zi Wang, Guangshuo Ou** *

Tsinghua-Peking Center for Life Sciences, Beijing Frontier Research Center for Biological Structure, McGovern Institute for Brain Research, State Key Laboratory of Membrane Biology, School of Life Sciences and MOE Key Laboratory for Protein Science, Tsinghua University, Beijing, China

* guangshuoou@tsinghua.edu.cn

**Data Availability Statement:** Mutation data of MMP dataset were directly downloaded from the MMP homepage (http://genome.sfu.ca/mmp/mmp_mut_strains_data_Mar14.txt).Python scripts

## Abstract

Chemical mutagenesis-driven forward genetic screens are pivotal in unveiling gene functions, yet identifying causal mutations behind phenotypes remains laborious, hindering their high-throughput application. Here, we reveal a non-uniform mutation rate caused by Ethyl Methane Sulfonate (EMS) mutagenesis in the *C. elegans* genome, indicating that mutation frequency is influenced by proximate sequence context and chromatin status. Leveraging these factors, we developed a machine learning enhanced pipeline to create a comprehensive EMS mutagenesis probability map for the *C. elegans* genome. This map operates on the principle that causative mutations are enriched in genetic screens targeting specific phenotypes among random mutations. Applying this map to Whole Genome Sequencing (WGS) data of genetic suppressors that rescue a *C. elegans* ciliary kinesin mutant, we successfully pinpointed causal mutations without generating recombinant inbred lines. This method can be adapted in other species, offering a scalable approach for identifying causal genes and revitalizing the effectiveness of forward genetic screens.

## Author summary

Exploring gene functions through chemical mutagenesis-driven genetic screens is pivotal, yet the cumbersome task of identifying causative mutations remains a bottleneck, limiting their high-throughput potential. In this investigation, we uncovered a non-uniform mutation pattern induced by Ethyl Methane Sulfonate (EMS) mutagenesis in the *C. elegans* genome, highlighting the influence of proximate sequence context and chromatin status on mutation frequency. Leveraging these insights, we engineered a machine learning enhanced pipeline to construct a comprehensive EMS mutagenesis probability map for the *C. elegans* genome. This map operates on the principle that causative mutations are selectively enriched in genetic screens targeting specific phenotypes amid the backdrop of random mutations.

and the model are accessible on GihHub at https://github.com/young55775/Genetorch-developing. C. elegans CHIP-chip and CHIP-seq data can be obtained from the modENCODE project (https://compbio.med.harvard.edu/modencode/webpage/Chromatin.v0.6.html) and visualized in Wormbase Genome Browser (JBrowse (wormbase.org)). ModEncode data can also be downloaded from GEO Accession Viewer (https://www.ncbi.nlm.nih.gov/geo/query/acc.cgi) with the accession number listed in S1 Table. Code availability: https://github.com/young55775/EMSForest-A-Machine-Learning-Enhanced-EMS-Mutagenesis-Probability-Map.

**Funding:** This work was supported by the National Natural Science Foundation of China Grants to GO (31991190, 31730052, 31861143042, 31525015, 31561130153) and National Key R&D Program of China Grants (2017YFA0503501, 2019YFA0508401). The funder had no role in study design, data collection and analysis, decision to publish, or preparation of the manuscript.

**Competing interests:** The authors have declared that no competing interests exist.

Applying this mapping tool to Whole Genome Sequencing (WGS) data derived from genetic suppressors rescuing a *C. elegans* ciliary kinesin mutant, we achieved precise identification of causal mutations without resorting to the conventional generation of recombinant inbred lines. Our work not only advances understanding of mutation dynamics but also revitalizes the efficacy of forward genetic screens, contributing to the refinement of genetic exploration methodologies with implications for various organisms.

## Introduction

Forward genetic screens have been instrumental in demystifying the molecular mechanisms underpinning myriad biological processes across a diverse array of organisms. The procedure commences with the creation of a genetically varied population, often achieved via the application of chemical mutagens such as Ethyl Methane Sulfonate (EMS) [1–3] or through irradiation [4,5], thereby seeding the genome with a variety of mutations. This random mutagenesis generates a pool of organisms each harboring a distinct set of genetic aberrations. Subsequent stages involve diligent phenotypic screening, identifying responsible genes, and decoding biological significance. This strategy has been pivotal in unveiling gene functions across multiple biological spectra, spanning developmental processes [6,7], signal transduction [8], and disease mechanisms [9]. Genes unearthed through these screens often illuminate deeper molecular mechanisms and are frequently employed to glean insights into analogous processes in more complex organisms, thereby contributing profoundly to our understanding of biological systems and phenomena.

Once a desirable mutant phenotype is identified through chemical mutagenesis, a key challenge lies in determining which among potentially thousands of induced mutations is responsible for the observed phenotype [10]. Identifying the causal mutation necessitates a substantial investment of labor and time, often demanding rigorous mapping and validation work to confirm the genetic change responsible for the observed phenotype [11–13]. These limitations, especially when contrasted with RNAi [14,15] or CRISPR-Cas9-based reverse genetics [16] methodologies, have led to the gradual obsolescence of EMS chemical mutagenesis and screens in the model organisms. RNAi, with its ability to knock down genes transiently and its amenability to high-throughput formats [17], enables the systematic analysis of gene function across the genome, offering a powerful tool for functional genomics [18]. With CRISPR-Cas9, not only can specific genes be targeted, but specific types of mutations (e.g., point mutations, deletions, insertions) can be introduced [19], offering a level of precision and control that is simply unattainable with chemical mutagenesis [20,21].

On the other hand, chemical mutagenesis has unique advantages in the generation of a broad spectrum of alleles [22], including hypomorphic, hypermorphic, and neomorphic alleles, providing a rich reservoir for investigating gene function in a depth unattainable through complete gene knockdown or knockout strategies like RNAi or CRISPR-Cas9. Thus, forward genetic screens facilitate the exploration of nuanced gene functions [23], interactions [24] and pathway dynamics [25], often unveiling surprising connections and uncharted biological landscapes that are not immediately evident through targeted gene perturbation approaches [26]. Furthermore, this strategy can illuminate the functionality of genes of unknown function or open reading frames that might be overlooked with hypothesis-driven reverse genetics [27]. This stochastic, phenotype-driven approach allows for the serendipitous discovery of novel genetic interactions and pathways without preconceived notions about gene function, offering the potential to uncover entirely new facets of biology.

Whole-genome sequencing (WGS) has emerged as a formidable instrument for pinpointing causal mutations derived from genetic screens [28]. In the context of *Caenorhabditis*

*elegans* (*C. elegans*), several strategies have been devised to reduce the number of candidate causal mutations. One prevalent method involves mating mutants, which are in the N2 background, with the polymorphic Hawaiian strain (CB4856), while alternative strategies exploit DNA variants inherent to the initial background or introduced through mutagenesis [29]. For the identification of suppressor mutations, the Sibling Subtraction Method (SSM) has been developed, which, by excluding genetic variants present in both mutants and their non-mutant siblings, markedly diminishes the roster of candidates [10]. However, these strategies, given their reliance on genetic crosses, do not lend themselves to high-throughput applications. Given that the recent advancement of AlphaMissense has predicted millions of pathogenic variants [30], the incorporation of homologous ones into model organisms for genetic suppressor studies may enhance our comprehension of rescue mechanisms and forge paths towards the proposal of innovative therapeutic strategies. Consequently, there is a pressing demand for the development of scalable methods for identifying causal mutations.

Machine learning, widely used as a statistical method, is a powerful tool for detecting hidden relationships between variables, fitting predictive models to data, and identifying informative groupings within data [31]. It proves particularly valuable when datasets are too complex for straightforward human analysis, enabling the capture of underlying relationships between variables. With rapid advancements in this field, successful attempts have been made to link phenotypes to potential causal variants based on sequencing data [32] or structural information [30]. Most existing tools predict the impact of a variant based on sequence conservation or clinical datasets, such as SIFT [33] or SnpEff [34]. However, when dealing with genetic screening involving highly effective mutagens, pathogenic alleles are not always the causal ones, and the genotype/phenotype relationship can be complicated by different genetic backgrounds. Furthermore, there are many genes whose functions are not fully understood and are not necessarily disease-causing but still worthy of study. A new model for genetic screening is needed to detect causal mutations independent of their known functions to predict the significance of mutated genes in chemical screening experiments.

In this study, we systematically analyzed *C. elegans* WGS data derived from 737 EMS-mutagenized worms, furnished by the Million Mutation Project (MMP) [22], alongside chromatin immunoprecipitation sequencing (ChIP-seq) datasets of DNA-binding proteins from the modENCODE project [35]. Contrary to expectations, we discovered that EMS-induced mutations were not uniformly distributed across the *C. elegans* genome. Instead, a genome-wide EMS mutagenesis bias was uncovered, which correlates with both adjacent sequence context and chromatin structure, facilitating the development of a machine learning based model for predicting EMS-induced mutagenesis probabilities. This model enabled the generation of a genome-wide EMS mutagenesis probability map, which predicts the expected frequency of random mutations for each nucleotide within the *C. elegans* genome. Consequently, our method enhances our capability to discern causative mutations—enriched through genetic screening for a specific phenotype—from random mutations. By employing this pipeline to analyze WGS data of genetic suppressors that rescue a ciliary kinesin harboring a missense mutation in *C. elegans*, we identified the causal mutations without the requisite establishment of recombinant inbred lines, thus enhancing the efficiency of high-throughput forward genetics.

## Result

### A non-uniform pattern of EMS-induced mutations across the *C. elegans* genome

We examined the distribution of a cumulative 265,169 Single Nucleotide Variants (SNVs) and insertion/deletions (InDels) across the 737 EMS-mutagenized genomes from the Million Mutation Project (Fig 1A). A conspicuous non-uniform pattern emerged, characterized by

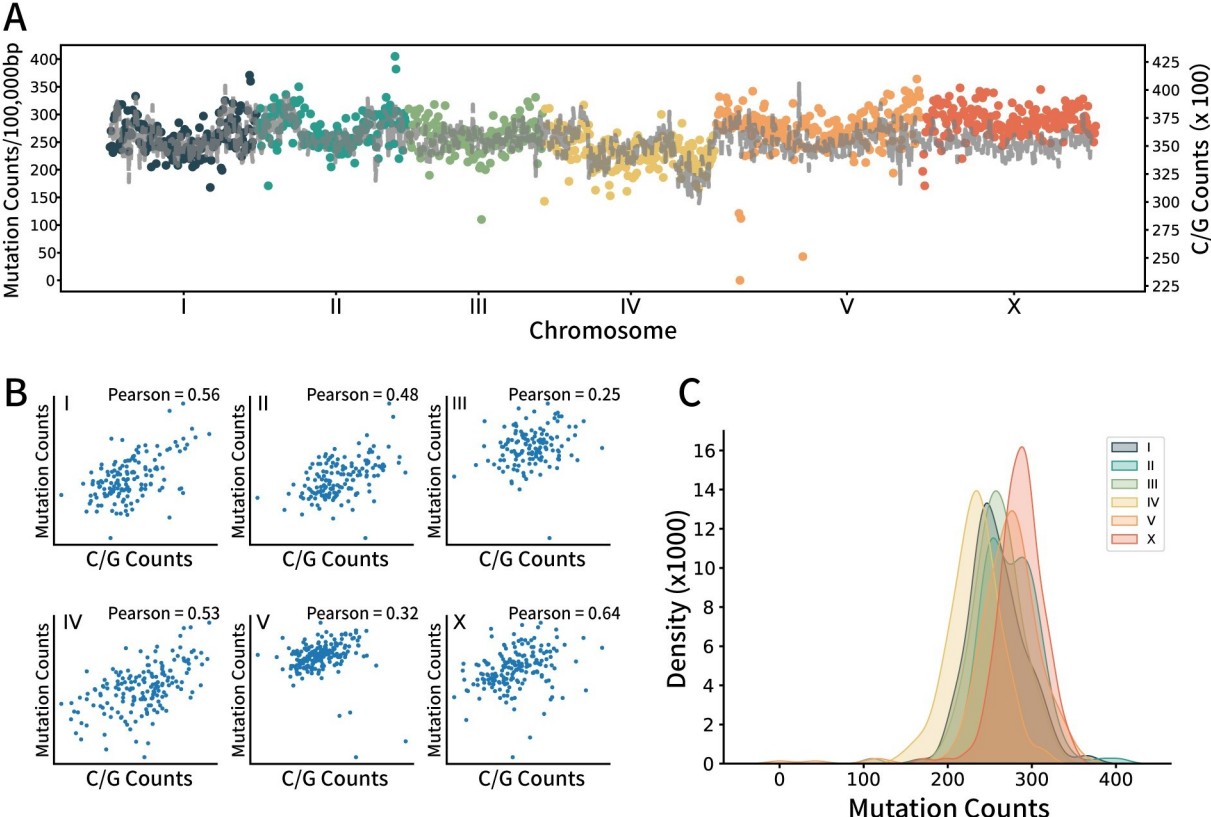

**Fig 1. Uneven distribution of EMS-induced SNV.** A) Scatter and line plots representing mutations in the MMP dataset. The Scatter plot shows the mutation number on each chromosome. Each dot represents the number of mutated bases within every 100,000 bp counted from the first base pair of each chromosome. The Line plot represents the number of 'C/G' base pairs in each 100,000 bp region. B) Scatter plot representing the relationship between 'C/G' base pair contents and the number of mutations from the MMP data of each chromosome. Each dot represents the number of mutated bases within every 100,000 bp counted from the first base pair of each chromosome. Pearson correlation was used to evaluate the association between them. C) Kernel density estimation plot of the number of mutations on each chromosome.

'hot spots'—regions exhibiting a higher-than-average SNV density—and 'cold spots,' or regions manifesting a lower density. Given that EMS predominantly introduces an alkyl group at N7-guanine and O6-guanine, consequently inducing transitions from 'G/C' to 'A/T,' we postulated that mutation frequency might be modulated by the genomic concentration of 'G/C' nucleotides. Nonetheless, our correlation analysis revealed that the 'G/C' pair distribution only partially elucidated the mutation frequency (Fig 1A and 1B). Moreover, we discerned variations in mutation frequency amongst different chromosomes (Fig 1C), with chromosome X exhibiting an elevated mutation frequency and chromosome IV manifesting a diminished frequency, neither of which could be attributed to the 'G/C' distribution. These observations parallel the findings from two additional WGS datasets procured from our recent Artificial Evolution Summer School, in which undergraduates perform forward genetic screens for animals exhibiting dumpy (Dpy) or uncoordinated movement (Unc) phenotypes (S1A–S1F Fig). These results imply that EMS-induced variations are not uniformly distributed across the genome, potentially due to disparate local chromosomal features.

## The adjacent sequence context influences the probability of EMS mutagenesis

We investigated the distribution of the four nucleotides both upstream (3' +4/+3/+2/+1) and downstream (5' -4/-3/-2/-1) of mutated bases across the genome (Fig 2A and 2B). Should

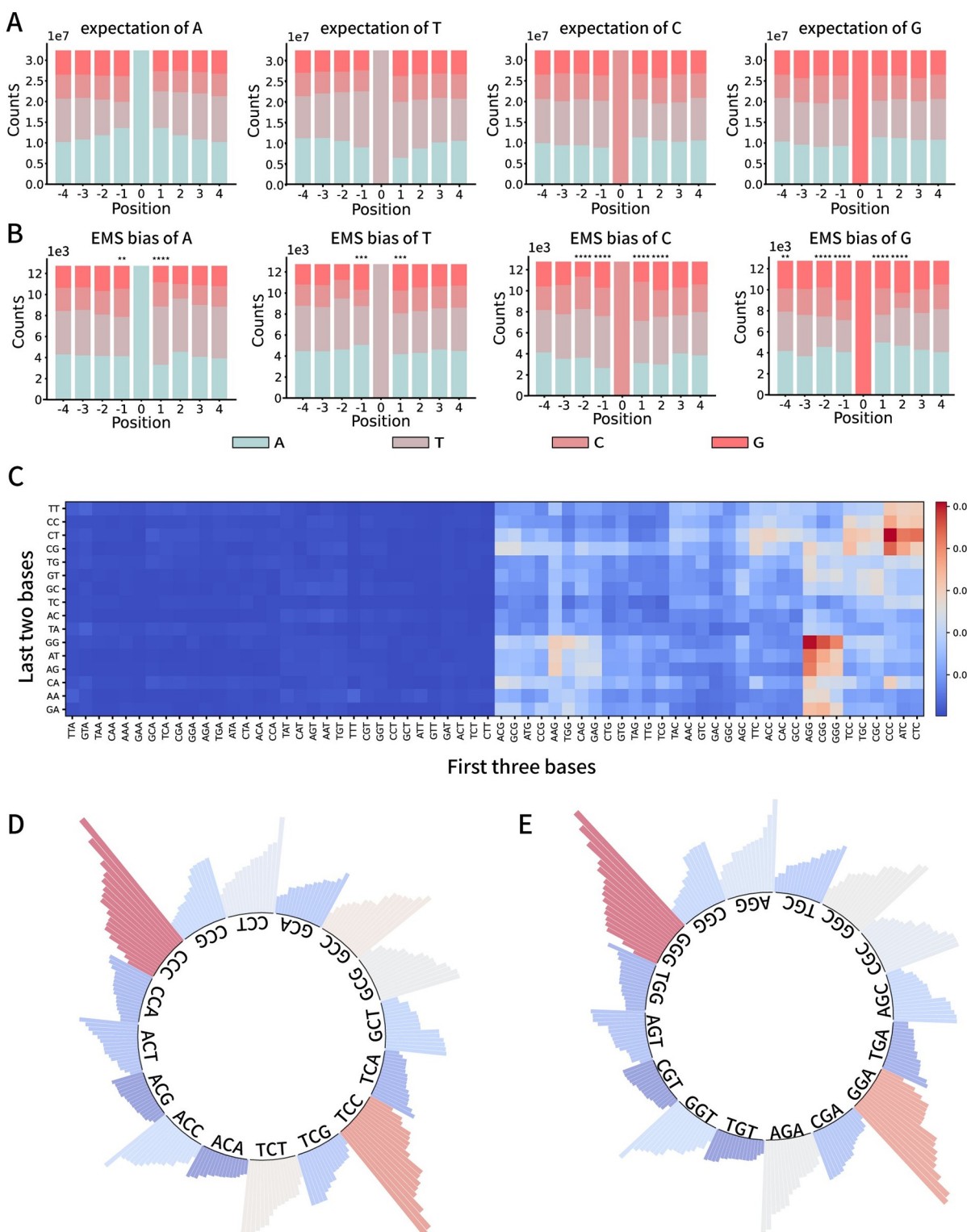

**Fig 2. Flanking sequences affect EMS mutagenesis efficiency.** A) The distribution patterns of flanking sequences of each kind of base (A, T, C, G) in the genome DNA. The number with '+' or '-' represents the base left (5')/right (3') to the position 0. These distributions were set as expectation values that if the mutagen selects randomly on a genome, it will cause mutations with the same distribution in flanking sequences. B) The distribution patterns of flanking sequences for each kind of mutated base in MMP dataset. Note that these distribution patterns are different from the corresponding distributions in (A). Chi-square test: * p<0.05, ** p<0.01, ***p<0.001, ****p<0.000. C) Heatmap recording

the mutation rate of each 5-base pattern. X-axis was sorted by the first 3 bases and Y-axis containing the next two. D) Bar plots showing the variation of C mutagenesis rate when -1 and +1 positions are fixed, as shown around the inner circle. E) Bar plots showing the variation of G mutagenesis rate when -1 and +1 positions are fixed, as shown around the inner circle.

EMS-induced mutations transpire indiscriminately, uninfluenced by adjacent bases, an asymmetry-free distribution would be anticipated. However, our chi-square test illuminated that the positions (+2/+1) and (-1/-2) adjacent to the 'C/G' nucleotide wield a significant impact on the efficacy of EMS mutagenesis. Examining the MMP dataset, we discerned that certain 5-base sequences exhibit a markedly heightened susceptibility to mutation. For example, sequences such as 'AGGGG' and 'CCCCT' are roughly tenfold more predisposed to mutation relative to sequences like 'GTCGA' and 'TCGAC' (Fig 2C). The substantive impact on mutagenesis probability is not confined to the immediate neighbors of nucleotides. Even while maintaining the +1/0/-1 position constant, the +2 and -2 positions continue to sway mutagenesis probability (Fig 2D and 2E), suggesting a viable avenue for predicting biases in mutation frequency. Utilizing this information, the base mutation frequency can be determined by calculating the mutation rate of each pattern in all the mutation events in the Million Mutation Project. Hereafter, we refer to this base mutation frequency as $P_0$, which is determined by: $P_0 = \frac{C'}{C_0}$. $C'$ represents the count of a specific five-base pattern underwent mutation in the MMP dataset and $C_0$ is the number of the same pattern in the reference genome.

## Multiple DNA-binding protein patterns show correlation with the mutation events

Employing the adjacent sequence context, we crafted a graphical representation that elucidates the frequency of EMS-induced mutations. To exemplify the method, we showcase the map in relation to mutation events across a broad range on chromosome V, spanning from position 1,800,000 to 3,450,000 (Fig 3A and 3B). We incorporated a representative gene, *ttn-1*, situated on chromosome V between positions 6,120,909 and 6,202,632 (S2A and S2B Fig). In both scenarios, the maps reveal expansive genomic segments characterized by a notable diminution in EMS-induced mutations, the 'silent region' of which belies the predictions formulated on adjacent sequence context.

We posited that a unique chromatin state might elucidate the emergence of these 'silent regions'. Tightly supercoiled DNA, potentially having circumscribed interactions with extraneous substances [36], along with particular DNA-binding proteins, might confer protection to their target sequences, rendering them less vulnerable to chemicals that engage with DNA [37]. Consequently, we probed the *C. elegans* ChIP-seq datasets of DNA-binding proteins and histone modifications from the modENCODE project (Figs 3C–3F and S2C–S2F). On chromosome V, a silent region extends over 200kb in length (Fig 3B). Within this expanse, the binding activity of the transcription factor UNC-62 and UNC-39 manifests irregularly, as evidenced by the ChIP-seq signal (Fig 3C and 3D). The absence of ChIP-seq signals in this area might denote a distinctive chromatin state that is less amenable to external substances. Analogously, the 'silent region' of the gene *ttn-1* also undergoes a comparable albeit less marked depletion of transcription factor binding (S2C and S2D Fig). Nonetheless, a distinct histone binding pattern is evident in this region (S2E and S2F Fig). While histone binding shows a slight increase across the preceding 200 kb 'silent region' on chromosome V (Fig 3E and 3F), particularly in H3, the profile of transcription factor binding provides a clearer boundary of this area. This suggests that transcription factor binding may be a more effective feature for capturing the relationship between abnormal mutation frequency and DNA-binding proteins

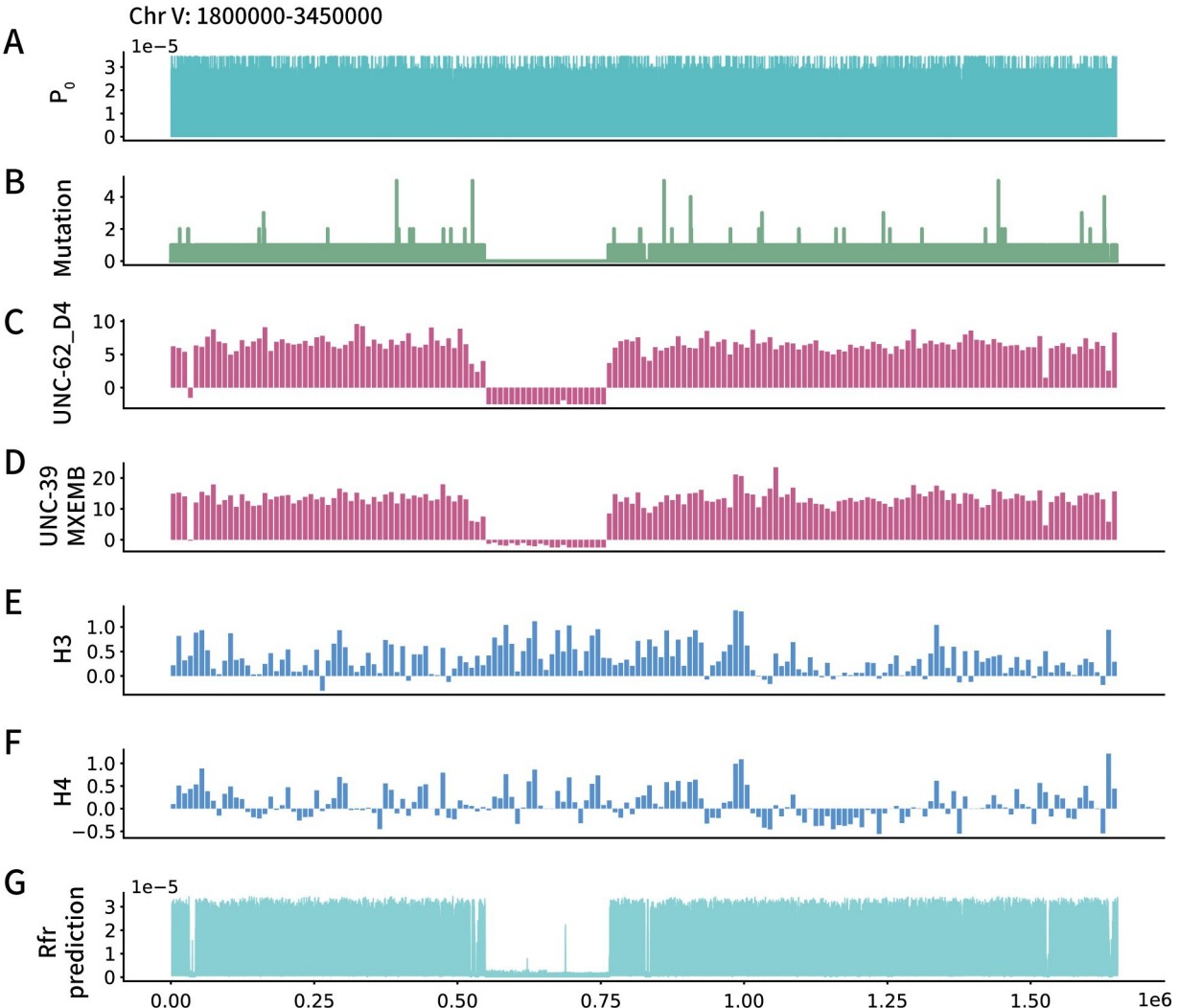

**Fig 3. Mutation numbers in the MMP data shows association with the DNA-binding protein features.** Take the MMP data and DNA-binding protein features on Chr V:1800000~3450000 as an example. A) Line plot showing the prediction made from the Flanking sequence preferences of EMS mutagenesis ($P_0$). B) The mutations observed in the MMP dataset. C) Raw ChIP-chip signal data of UNC-62 binding in young adult worms. D) Raw ChIP-chip signal data of UNC-39 binding in worm embryo. E) Raw ChIP-seq signal data of histone H3 in L3 larval worms. F) Raw ChIP-seq signal data of histone H4 in L3 larval worms. G) Line plot showing the prediction made by a Random Forest regressor trained by DNA-binding protein data.

in this case. Probing mutations and the attributes of DNA-binding proteins suggests that regions with elevated or reduced mutation probabilities may originate from direct causative factors or mere coincidence. Nevertheless, it seems judicious to elucidate the binding patterns of these proteins and explore their association with mutation probability.

Using ChIP-seq dataset collections from the WormBase, we next examined the potential correlation between DNA-binding protein features and mutagenesis probability with machine learning techniques.

Nevertheless, it seems judicious to elucidate the binding patterns of these proteins and explore their association with mutation probability. Employing ChIP-seq dataset compilations from WormBase, we subsequently scrutinized the potential correlation between DNA-binding protein characteristics and mutagenesis probability through the lens of machine learning methodologies.

## Random Forest regressor (Rfr) modeling to predict the mutation frequency

Employing ChIP-seq dataset compilations from modENCODE and WormBase [38], we examined the potential correlation between DNA-binding protein characteristics and mutagenesis probability through machine learning methods, which can identify complex chromatin feature patterns and learn their relationship with mutation events

A decision tree is a simple yet powerful prediction method [39]. It splits data into subsets in order to create groups with similar values of the target feature. Therefore, the predictor can make predictions from new observations with the relationship it learned from the already exist data. Random Forests are a combination of tree predictors [40]. Each tree is trained on a randomly selected subset of the data and features, capturing slightly different information, and reducing the risk of overfitting, thus improving overall performance.

We developed a regression model using Random Forest regressor. The underlying hypothesis is that, despite the sparsity of mutation events in the Million Mutation Project (200–350 mutations per 100,000 bp, Fig 1A), the mutation probability for base pairs with the same 'property' is well-represented in the dataset. The random forest algorithm's task is to categorize base pairs into different groups based on DNA binding protein patterns and flanking sequences, enabling accurate prediction of each group's average mutation expectation (Fig 4A). Nucleotides with similar flanking sequences and DNA-binding protein attributes are grouped into the same output node of each tree. The mean mutagenesis frequency for that node then serves as the predicted mutagenesis frequency for any nucleotide with similar characteristics assigned to that node during predictions.

To create our model, we used a dataset containing 24 DNA-binding protein datasets from modENCODE, which includes information about histone binding, modifications, transcription factor interactions, and other epigenetic modifications (S1 Table). We constructed 600 decision trees, each based on the mentioned method, with each tree trained on a randomly chosen subset of up to 16 protein features. To create our model, we used a dataset containing 24 DNA-binding protein datasets from modENCODE, which includes information about histone binding, modifications, transcription factor interactions, and other epigenetic modifications (S1 Table). each trained on a randomly chosen subset of up to 16 protein features, based on the method described above.

The ultimate model output can be succinctly represented as $P = \frac{\alpha}{600}\sum_{i=1}^{600} f_i(x, P_0)$ (see also Materials and Methods), which involves averaging the predictions from each individual tree to provide a more precise overall prediction. In this equation, the variable $\alpha$ factor in experimental variations stemming from fluctuations in EMS concentrations, developmental stages, ambient temperatures, and other potential elements that may impact EMS mutagenesis effectiveness.

To validate the model's performance, our initial validation assessed predictions grounded solely on $P_0$, demonstrating congruent patterns across every 300 kb genomic block. These patterns were juxtaposed with the discerned fluctuations in mutation occurrences within the MMP dataset (Fig 4B). Upon training utilizing a randomly selected 80% of the data, the model's performance was appraised against the test set. The test set was categorized based on prediction results and subsequently partitioned into 100kb blocks. Anticipated mutation tallies were then contrasted against actual mutation instances documented in the MMP dataset (Fig 4C). Further, a thorough evaluation juxtaposed the anticipated mutation tallies within 300kb genomic blocks (Fig 4D) and individual genes, as per the WBcel235 annotation (Fig 4E), against actual mutation tallies from the MMP dataset. Collectively, these comparisons refine the rudimentary $P_0$ map, facilitating a meticulous prediction of EMS mutagenesis probability at individual nucleotides (Figs 3G, 4F, 4G and S2G).

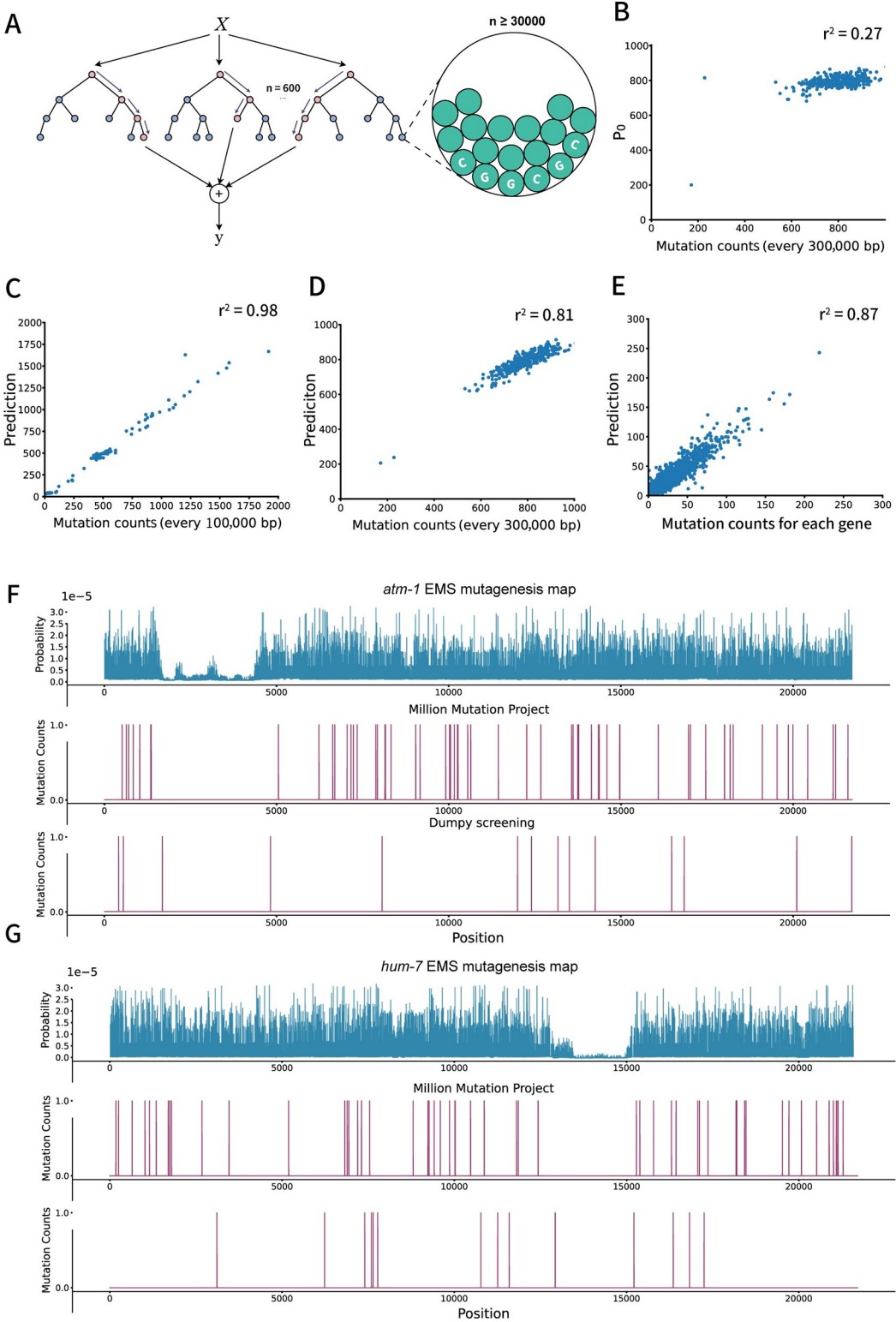

**Fig 4. Random Forest modeling and validation.** A) Schematic diagram of the Random Forest regressor (RFr) model. 600 trees were modeled and each tree randomly used at most 70% percent of the 24 features (≤16 features). The smallest leaf size is set to be 30000 to avoid overfitting. In this model, each decision tree in the forest will put bases with similar properties (flanking sequences and DNA-binding protein patterns) together into an output node, and use the average mutation rate in each node as output. Consequently, the forest will take the average output of every 600 trees as the final

output, with which the mutation rate of this kind of base can be predicted. B) Prediction made only by $P_0$. The worm genome was divided into 300,000 bp blocks and each dot represents the expectation of mutation number made by $P_0$ and the actual mutation counts in MMP dataset. C) The performance of RFr on the test set, which contains 20% of the whole dataset. The test set was sorted by the prediction and was divided into 100,000 bp blocks. Each dot represents the expectation of mutation number made by RFr and versus actual mutations counts in MMP dataset in each 100,000 bp long blocks. D) Scatter plot of the overall performance of RFr on the whole genome. The genome was divided into 300,000 bp blocks. Each dot represents the expectation of mutation number made by RFr and the actual mutations counts in MMP dataset in each 300,000 bp long blocks. E) Scatter plot of the overall performance of RFr on 19639 genes of *C. elegans*. Each dot represents the expectation of mutation number made by RFr and the actual mutations counts in MMP dataset of each gene. F) Representative EMS mutagenesis map of gene *atm-1*. Followed by the mutation map in the MMP dataset and dumpy screening. G) Representative EMS mutagenesis map of gene *hum-7*. Followed by the mutation map in the MMP dataset and dumpy screening.

Subsequently, we selected two representative genes and created EMS mutagenesis maps, highlighting variations in mutagenesis frequency across the entire gene. These maps were then compared to mutations observed in the MMP dataset and sequencing data from the dumpy screening mentioned earlier (Fig 4F and 4G). In regions identified as 'silent region' based on our predictions, neither screening approach revealed mutations. Conversely, due to the limited number of SNVs in the MMP dataset, numerous nucleotides remained unmutated in this data-set. Notably, the continuous absence of observed mutations on the actual mutation occurrence map did not diminish the model's ability to predict the potential for mutagenesis: In the dumpy screening, we observed mutations in regions where the MMP dataset displayed mini-mal mutations (Fig 4F and 4G). This suggests that our model recognized the specific character-istics of nucleotides in these regions and transferred the knowledge that similar nucleotides were prone to mutation in other genomic regions in order to make accurate predictions.

## The EMS mutagenesis probability map facilitates the identification of causal mutations

We employed the EMS mutagenesis probability map to discern the causal mutation within a genetic suppressor screen, amending defects instigated by a missense mutation in the ciliary kinesin OSM-3. This kinesin drives intraflagellar transport, a process pivotal for the construc-tion of olfactory cilia in *C. elegans* sensory neurons. These segments harbor an abundance of G protein-coupled receptors [41], enabling the organism to perceive environmental stimuli, inclusive of various odorant molecules. A functional impairment of the OSM-3 kinesin culmi-nates in a specific loss of distal ciliary segments [42]; concomitantly, the delocalization of GPCRs or the prevention of odorant-receptor interaction causes animal behavioral defects, such as an incapacitation in executing osmotic avoidance (Osm).

Our preceding genetic screens isolated the E251K missense mutation within the motor domain of OSM-3 kinesin. This mutation parallels a pathogenic variant, E253K, identified in the KIF1A kinesin [43], mutations within which give rise to a spectrum of neurological disor-ders collectively recognized as KIF1A-Associated Neurological Disorder (KAND). The E253K mutation in KIF1A is hypothesized to disturb the flexibility of switch I in the motor domain, thereby suppressing γ-phosphate release and subsequent ATP binding and hydrolysis within the motor domain [44,45]. An in vitro single-molecule motility assay revealed that E253K induces a stringent binding to the microtubule (MT) yet precludes the engagement in proces-sive motion of KIF1A. Consistently, *C. elegans* harboring the E251K mutation in OSM-3 dis-rupts their distal ciliary segments with full penetrance. Whereas wild-type animals exploit their distal ciliary segments to uptake the fluorescent dye DiI from the culture medium [46], all examined OSM-3 E251K mutant animals failed to do so (N > 500), manifesting a dye filling

defect (Dyf) phenotype unable to take up dye from the environment, which is detectable under a fluorescence stereoscope.

Utilizing the Dyf phenotype as an efficacious readout probing ciliary defects, we executed genetic suppressor screens aimed at restoring dye-filling capacity and recuperating ciliary distal segments. Our screens isolated 38 independent suppressors. We conducted whole-genome sequencing of all suppressors and employed the EMS mutagenesis probability map to analyze the WGS data. Initially, background mutations, defined as those shared across a wide range of samples, were identified, and removed. Subsequently, we evaluated the effectiveness of EMS mutagenesis (Materials and Methods). The remaining mutations were compared to the adjusted mutagenesis map (Fig 5A). To assess the enrichment effect on each gene by the genetic screening, we calculated the fold change and p-value for each gene showing mutations in the background-removed mutation pool (Fig 5A).

Our analysis unveiled the top five candidate genes, which demonstrated a markedly biased EMS mutagenesis rates in our suppressor screen but did not exhibit any enriched mutation frequency in our Dpy or Unc screens (Fig 5B). Among these candidates, two ciliary kinases, DYF-5 and DYF-18, have been recognized for their efficacy in rescuing ciliary defects in *osm-3* mutant animals. We obtained 10 and 16 mutant alleles for *dyf-5* and *dyf-18*, respectively. The majority of these mutant alleles induce missense mutations, introduce stop codons, or cause splicing mutations within the coding region. These results underscore that the EMS mutagenesis probability map facilitates the identification of causal mutations in genes known to participate in this process. The remaining three genes are not well characterized, and it is unclear which one or ones might be implicated in the regulatory mechanisms of OSM-3.

Through categorizing the mutational characteristics of three genes, we observed that two of them harbored mutations—other than missense, introduced stop, or splicing mutations—which are unlikely to disrupt the function of the gene products. In stark contrast, 12 genetic suppressors encompass various loss-of-function mutations within the coding region of the K04F10.2 gene. Noteworthily, given the substantial size of the initial group subjected to screening, the concurrence of two suppressor alleles within the same strain emerges as exceptionally rare. Notably, the mutations with a significant impact did not coincide across the 38 suppressor strains (Fig 5D), suggesting that the 38 genetic suppressors induce defects in three genes: *dyf-5*, *dyf-18*, and K04F10.2. Given that we have already procured 12 distinct alleles of K04F10.2, and generating additional mutants of this gene may not yield further insight, we endeavored to conduct transgenic rescue experiments to further determine whether K04F10.2 operates as the suppressor gene. To this end, we introduced the genomic DNA of K04F10.2, tagged with the red fluorescent protein Scarlet and controlled by its 2kb endogenous promoter, into OSM-3 E251K; K04F10.2 double mutant animals (Fig 5E and 5F). The IFT-dynein heavy chain CHE-3, which traverses the entire length of cilia, was marked with green fluorescence, and used as a ciliary marker to ascertain ciliary length (Fig 5E) [41]. As anticipated, GFP fluorescence was observed along cilia measuring 8.09±0.86 μm. In mutants harboring the OSM-3 E251K mutation, the distal ciliary segment was absent, resulting in cilia of the shorter length (4.40±0.93 μm). While OSM-3 E251K;K04F10.2 double mutants restored ciliary length to 7.54 ±0.60 μm, introducing Scarlet-tagged wild-type genomic DNA of K04F10.2 into the double mutant reduced ciliary length to 6.22±0.81 μm, similar to that of the OSM-3 E251K single mutant. These results indicate that K04F10.2 mutations act as suppressors for OSM-3 E251K. Henceforth, we designate this gene as Joubert syndrome homolog 26 (*jbts-26*) due to its inferred role in primary cilia. These results demonstrated that the EMS mutagenesis probability map is instrumental in isolating causal mutations in genes previously unlinked to specific processes.

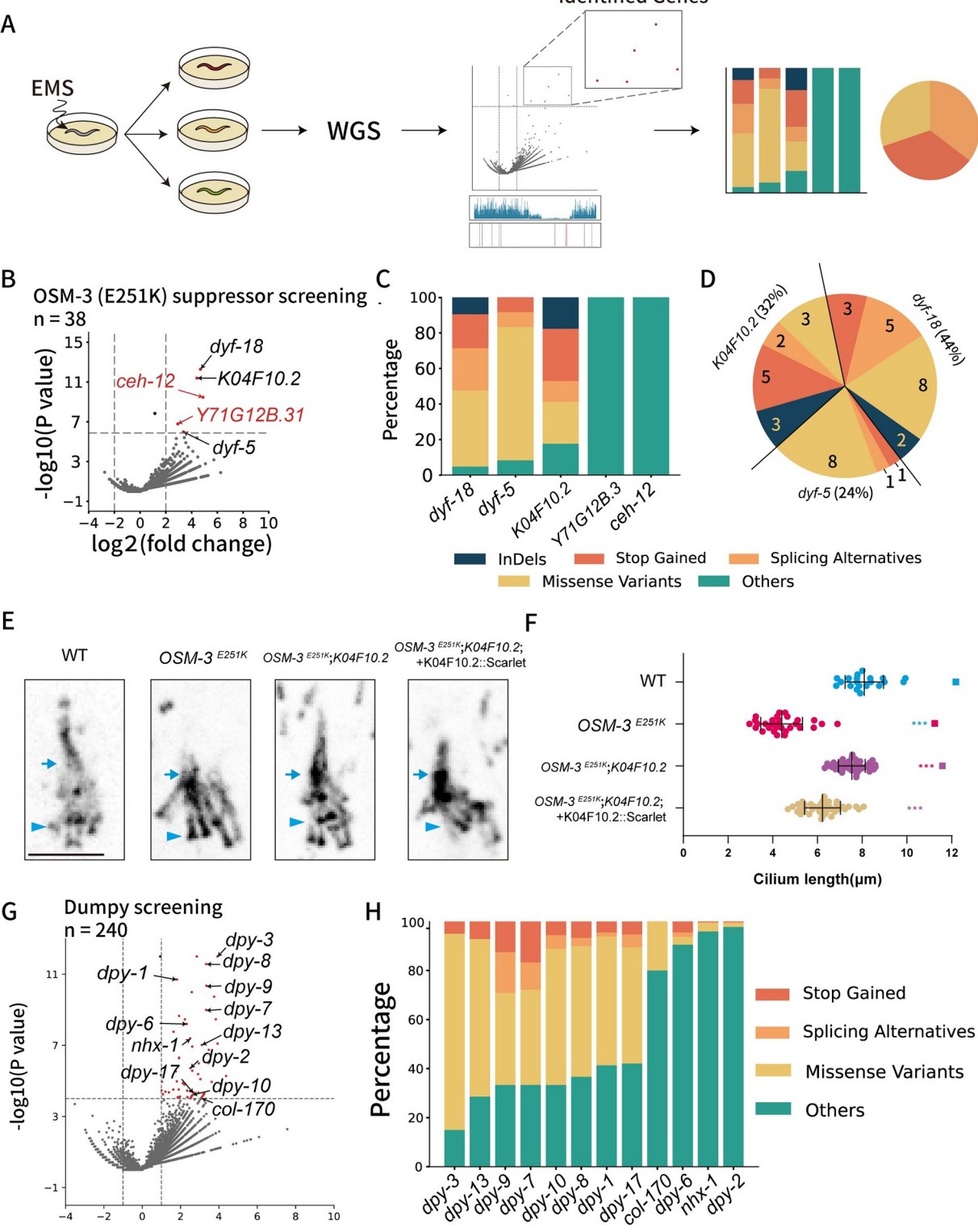

**Fig 5. Predict target genes with the EMS mutagenesis map.** A) Schematic diagram of the analysis pipeline. After genetic screening of the EMS mutated worms, those with a target phenotype were sequenced and mutation information was pooled. Then it could be compared with the EMS mutagenesis map based on the RFr (see Materials and Methods). Finally, candidate genes' mutation profiles were analyzed by the loss of function impact of InDel, stop-gain, splicing, missense mutations. B) Volcano plot showing the result of an OSM-3 (E251K, cas22599, n = 38) suppressor screening comparing to the EMS mutagenesis map. There were 5 genes showing significant difference between mutation expectation

and actual mutation number. C) Bar plot showing the mutation properties of these five genes. D) Taking all the mutations inducing a high impact on protein into consideration, each of the 38 samples has one high-impact mutation in one of the three genes. E) Ciliary defects in the OSM-3 (E251K, cas22599) mutant animals rescued by a K04F10.2 mutated allele (Arg506*, cas23441). K04F10.2 gDNA tagged with Scarlet overexpression in cas23441 exhibits shorter cilia. Arrowheads indicate the ciliary base and transition zone, and arrows indicate junctions between the middle and distal segments. F) Cilium length (mean ± SD) in each group, n = 30 to 50. *** P<0.001 one-way ANOVA. G) Volcano plot showing the result of a dumpy screening (N = 240) from EMS mutated N2 wildtype strain comparing to the EMS mutagenesis map. Dumpy-related or body wall related genes are labeled. H) Bar plot showing the mutation properties of genes labeled in (G).

## Discussion

In conclusion, this research delineates the non-uniform mutation rate instigated by EMS mutagenesis across the *C. elegans* genome, underscoring that both proximate sequence context and chromatin status are ostensibly correlated with mutation frequency. Leveraging these factors, we deployed a Machine Learning assisted pipeline to sculpt a genome-wide EMS mutagenesis probability map. Operating under the premise that causative mutations will achieve enrichment through genetic screens pertinent to a specific phenotype amidst random mutations, we utilized the map to scrutinize Whole Genome Sequencing (WGS) data derived from *C. elegans* forward genetic screens. The findings illuminate that the map expediently facilitates the discernment of genes known for their instrumental roles in these processes. Notably, the map also prognosticated a novel gene entwined in each regulation, a prediction substantiated through transgenic rescue experiments. This holistic approach to causal gene identification eschews labor intensive genetic crossing and demonstrates that bioinformatic analysis of WGS data via the EMS mutagenesis probability map can present a potent conduit for directly interfacing phenotype with genotype. Our model performs well when multiple alleles per gene are obtained but is less effective when screens yield only one or two alleles per gene. For challenging forward genetic screens such as manual fluorescence-based screens under high-power microscopy, we recommend using the sibling subtraction method. Machine learning approaches rely on statistical effects to achieve desired accuracy, assuming a sufficient number of candidates are screened—typically, we recommend 30 or more samples per experiment. Our tool was originally designed for large-scale causal gene identification. For small datasets, sibling subtraction remains the most reliable method due to its manageable workload when dealing with only a few alleles.

Given the affordability of WGS, WGS data are usually available before sibling subtraction experiments. Analyzing one hundred WGS datasets using the EMS probability map takes less than one minute, and only several seconds for 10 to 20 WGS datasets. Our model significantly enhances the likelihood of identifying potential causal mutations for genes that have low baseline mutation probabilities, irrespective of the number of isolated mutant alleles. Therefore, investing an additional minute in computation could potentially expedite mutant cloning efforts.

While the EMS probability map cannot be universally applied, especially in challenging screens that yield a limited number of mutant alleles, it proves valuable in many forward genetic screens capable of efficiently generating dozens of mutants with manageable effort. Nonetheless, it's crucial to employ multiple complementary approaches to enhance forward genetic screens. Their strength lies in their capacity to uncover novel mutations that might otherwise go unnoticed, and ability to unveil unknown aspects of biology. Thus, the EMS probability map can be a useful addition to the forward genetic toolkit.

The Random Forest model elucidated the relationship between various DNA-binding protein patterns and mutation frequency. DNA-binding proteins such as histones and transcription factors are commonly employed to assess the accessibility of genomic DNA, from which the topological arrangement of nucleosomes can be inferred [47]. However, this relationship

becomes more intricate due to the challenge of precisely uncovering how DNA-binding protein patterns affect EMS effectiveness. For example, the occupancy of transcription factors on chromatin may reduce accessibility to EMS, while regions lacking ChIP-seq signals likely indicate a tightly coiled DNA topology with nucleosomes, making them less accessible to other molecules. The machine learning model established the relationship between EMS mutation frequency and these features by leveraging these correlations, resulting in a comprehensive model capable of handling diverse patterns. Further experiments are necessary to decipher these patterns and gain a deeper understanding of the actual topology and conditions of these regions.

We summarize the mutation counts from various datasets: In the MMP project, 737 WGS datasets of EMS mutagenized worms revealed a total of 265,169 mutations, averaging 360 mutations per worm, or approximately 0.36 mutations per 100,000 base pairs (bp). The DPY dataset, consisting of 240 worms, showed 65,171 mutations, averaging 272 mutations per worm, or approximately 0.27 mutations per 100,000 bp. Lastly, the UNC dataset, with 118 worms, yielded 34,730 mutations, averaging 293 mutations per worm, or approximately 0.30 mutations per 100,000 bp. While the MMP project generates a slightly higher mutation rate (360 mutations per worm) compared to our DPY (272 mutations per worm) or UNC (293 mutations per worm) screens, all mutation rates fall within a similar range.

The elucidation of *jbts-26* as a suppressor gene mitigating ciliary defects induced by the OSM-3 E251K mutation inaugurates avenues for mechanistic investigations. Previous research has illuminated that the abrogation of ciliary kinases DYF-5 or DYF-18 permits an alternative ciliary heterotrimeric kinesin-2 to ectopically enter the ciliary distal region [48], thereby substituting the functionality absent in OSM-3. In essence, these two ciliary kinases may not exert their influence directly upon OSM-3. Rather, they curtail a concurrent ciliary transport pathway, and their loss enables heterotrimeric kinesin-2 to compensate for the OSM-3 deficit. Consequently, in the singular mutants of *dyf-5* or *dyf-18*, the organisms exhibit aberrantly elongated cilia [48]. In contrast, *jbts-26* might orchestrate its function through a mechanism disparate from these two ciliary kinases. We and others have not been able to discern any overt ciliary defects in the single mutant defective in *jbts-26*. A pioneering systematic exploration of ciliary genes has annotated *jbts-26* as a conserved putative binding associate of a microtubule-severing protein, namely katanin, hinting at a role in microtubule regulation [49]. Nonetheless, the underlying mechanisms remain mysterious. Considering that the E251K mutation in OSM-3 is analogous to the E253K mutation in KIF1A, future studies will ascertain whether the inhibition of *jbts-26* can ameliorate neuronal defects induced by KIF1A E253K, potentially unveiling a novel therapeutic target for intervening in KIF1A E253K-associated neurological disorders.

Recent endeavors to conduct genetic suppressor screens for various missense mutations within OSM-3 have been undertaken, inclusive of published suppressors of a motor hinge mutation, OSM-3 G444E [44,45,50]. Nonetheless, no mutations of *jbts-26* were unveiled as suppressors for the ciliary defects attributed to OSM-3 G444E, positing that *jbts-26* might exert its suppressive effects on the E251K mutation in OSM-3 in a residue-specific modality. This observation accentuates the imperative of functional residuomics: each residue may exhibit intrinsic uniqueness, and mutations at each individual residue may be ameliorated via divergent, distinct mechanisms.

Given the ubiquitous chemical principle underpinning EMS's capacity to induce genetic mutations across diverse species, it is conceivable that the adjacent sequence context and chromatin status might also engender a non-uniform distribution of EMS-induced alterations throughout all genomes. Congruent with our findings, the efficacy of EMS mutagenesis in rice (*Huanghuazhan*) is correlated with its flanking sequence and chromatin status [51].

Consequently, it may not be startling to observe that systematic analyses of published WGS data across species divulge a pervasive bias in EMS mutation rates at genome-wide levels. Identifying EMS-induced causal mutations in species characterized by complexities in genome size, ploidy number, and life cycle surpassing those of *C. elegans* poses a substantially more formidable challenge. Therefore, formulating analogous EMS mutagenesis probability maps for these species could markedly expedite forward genetics endeavors therein.

## Materials and methods

### WGS data from the Million Mutation project

WGS data from the Million Mutation project can be accessed via the official webpage of Simon Fraser University (http://genome.sfu.ca/mmp/). For our study, we utilized mutation data from a total of 737 strains isolated after EMS mutagenesis was used to train and test the model. Raw sequencing data in fastq format are available from Home—SRA—NCBI (nih.gov). The average sequencing depth of these data were about 15x fold and 265169 variations were found in this EMS mutation dataset.

### DNA-binding protein data

*C. elegans* ChIP-chip and ChIP-seq data can be obtained from the modENCODE project (data.modencode.org) and visualized in WormBase Genome Browser (JBrowse (wormbase.org)).

### $P_0$ calculation from the MMP dataset

Flanking sequence preferences of EMS mutagenesis (referred to as $P_0$) were calculated with the MMP dataset and *C. elegans* genome WBcel235 (*Caenorhabditis_elegans* - Ensembl Genomes 57) as

$$P_0 = \frac{C'}{C_0}$$

$C'$ represents the count of a specific five-base pattern underwent mutation in the MMP dataset and $C_0$ is the number of the same pattern in the reference genome.

### Preparation of DNA-binding protein data and Random Forest regressor modeling

The raw ChIP-chip and ChIP-seq data were preprocessed to better reflect the actual distribution of DNA-binding protein on the genome. A python script was employed to apply a moving average with the window size of 100 bp to smooth the data across each chromosome. The smoothed DNA-binding protein dataset was then utilized to train and validate a random forest regressor after randomly spilt the data into a 80% training set and a 20% test set.

The random forest comprised a total of 600 decision trees, each trained on the provided training dataset. To prevent overfitting, we implemented feature selection by randomly considering up to 70% of the available features for each tree. Additionally, to maintain model integrity, we set a minimum leaf size of 30,000 instances. To produce the final prediction, the output from each individual tree was aggregated. This ensemble approach allowed us to generate a comprehensive mutation probability assessment for each type of nucleotide.

During validation, the test set, comprising both mutation information and additional features, was sorted based on their predicted mutation rates. Subsequently, the test set was divided into 100kb blocks. To validate the model's predictions, we computed the mutation

expectation and the actual mutation count in the MMP dataset. By calculating the likelihood of mutation for each individual base pair, we were able to determine the mutation rate for each block, or for any genomic sequence length, using the formula:

$$P_{seq} = \sum\nolimits_{i \in seq} P_i.$$

This approach allowed us to assess the model's predictive accuracy across different genomic regions.

During a genetic screening involving the potential mutation of millions of base pairs, the number of variations for each nucleotide can be reasonably modeled to follow a binomial distribution. Consequently, the expected mutation count can be calculated as:

$$E(seq) = \sum\nolimits_{n \in seq} E(n) = n * \sum\nolimits_{n \in seq} P_n.$$

This formula is applicable within a population of n strains, and it allows us to estimate the anticipated number of mutations across the sequence under investigation.

The EMS effectiveness can be subject to variation due to factor such as changes in temperature, the age of the worms, minor temporal differences, and alterations in mutagen concentration during the EMS mutagenesis process. To assess this effectiveness, we compare the average number of mutation events per strain in different batches of EMS mutation experiments. We took the mutation frequency of the Million Mutation Project as a baseline. We denote the mutagen effectiveness of the MMP strains as $\alpha_0$ and the efficiency in another experiment as $\alpha_1$, the random forest regression model takes the form of:

$$P = \alpha_0 f(x, P_0)$$

where x represents the DNA-binding protein features used by the model to make predictions and adjust the initial prediction of $P_0$. This model can be applied to another dataset as:

$$P\prime_{seq} = \alpha_1 \sum\nolimits_{i \in seq} f(x_i, P_0) = \frac{\bar{N}_1}{\bar{N}_0} \alpha_0 \sum\nolimits_{i \in seq} f(x_i, P_0)$$

Here, $\bar{N}_0$ and $\bar{N}_1$ represent the average number of mutation events that occurred in the MMP strains and the batch to be analyzed, respectively. In this way, this model can be used on screening with altered EMS effectiveness.

## Model evaluation and feature importance

To evaluate the accuracy of this regression model, we compare the expectation and the actual mutation count within different regions of the *C. elegans* genome, resulting in the determination of the coefficient of determination ($R^2$), given by:

$$R^2 = 1 - \frac{MSE}{Var}$$

Additionally, we assess the importance of each feature used in making predictions through permutation scores. In this analysis, each feature is successively replaced with random noise sharing the same value distribution as the original data. The model, initially trained on the original dataset, is then employed to make predictions using the dataset in which a feature has been replaced with noise. The permutation importance of feature *j* is subsequently calculated

as:

$$i_j = R^2 - \frac{1}{n}\sum\nolimits_{i=1}^{n} R_{i,j}^2$$

Where feature $j$ undergoes $n$ shuffling iterations.

The mutation and feature importance data (S3 Fig), have been normalized to a range of 0 to 1 using the formula:

$$V = \frac{v - \min(v)}{\max(v) - \min(v)}$$

### Worm culture

*C. elegans* were maintained under a consistent temperature of 20°C according to the standard method. Nematode growth medium (NGM) with *Escherichia Coli* OP50 seeded on it was used to cultivate these worms. *C. elegans* strains used in this study are listed in S2 Table.

### EMS mutagenesis

Worms synchronized at the late L4 stage were carefully collected with 4 mL M9 buffer (S1 Table). Subsequently, these collected worms were placed in 50 mM EMS buffer at room temperature with continuous rotation for a duration of 4 hours. Following this treatment, the worms underwent a thorough washing process with M9 buffer and were then cultured under standard conditions. Approximately 20 hours later, the adult worms were subjected to a bleaching procedure to isolate their eggs (referred to as F1). These eggs were subsequently distributed across approximately 100 separate 9 cm NGM plates, with an average of 50 to 100 eggs placed on each plate. Any adult worms displaying the desired phenotype were meticulously collected and placed in individual culture settings. After a careful examination of their offspring, these individuals were subjected to sequencing using an Illumina next-generation sequencer.

### WGS data analyze

Mutation data of MMP dataset were directly downloaded from the MMP homepage (http://genome.sfu.ca/mmp/mmp_mut_strains_data_Mar14.txt). Raw reads obtained from the next-generation WGS were assessed for duplication and quality with FastQC and were trimmed using Trim_galore (version 0.4.4) to remove the adaptor sequence and low-quality reads. After that, clean reads were aligned to the reference genome (WBcel235) using BWA-MEM2 (version 2.2) with default parameters. In all these sequencing results, > 20x average coverage is ensured. Variations were detected using freebayes (version 1.3.6) and annotated using SnpEff. Finally, a filter was set to ensure the variation quality and limit false positive rate: only sequence depth >5 and allele frequency > 0.8 variations were taken into further analyze.

### Strain construction

To perform the rescue experiment, we synthesized the pK04F10.2::K04F10.2::Scarlet DNA fragment using SOEing PCR [47] and subsequently created the transgenic strain through microinjection.

### Supporting information

**S1 Fig. Uneven distribution of EMS-induced SNV in screening of dumpy and uncoordinated phenotype.** A) Scatter and line plots representing mutations in the Dumpy screening

dataset (n = 240). The Scatter plot shows the mutation number on each chromosome. Each dot represents the number of mutated bases within every 100,000 bp counted from the first base pair of each chromosome. The Line plot represents the number of 'C/G' base pairs in each 100,000 bp regions. B) Scatter plot representing the relationship between 'C/G' base pair contents and the number of mutations from the Uncoordinated screening data of each chromosome. Each dot represents the number of mutated bases within every 100,000 bp counted from the first base pair of each chromosome. Pearson correlation was used to evaluate the association between them. C) Kernel density estimation plot of the number of mutations on each chromosome in uncoordinated screening dataset. D) Scatter and line plots representing mutations in the Uncoordinated screening dataset (n = 118). The Scatter plot shows the mutation number on each chromosome. Each dot represents the number of mutated bases within every 100,000 bp counted from the first base pair of each chromosome. The Line plot represents the number of 'C/G' base pairs in each 100,000 bp regions. E) Scatter plot representing the relationship between 'C/G' base pair contents and the number of mutations from the dumpy screening data of each chromosome. Each dot represents the number of mutated bases within every 100,000 bp counted from the first base pair of each chromosome. Pearson correlation was used to evaluate the association between them. F) Kernel density estimation plot of the number of mutations on each chromosome in dumpy screening dataset.
(TIF)

**S2 Fig. Mutation numbers in the MMP data shown association with the DNA-binding protein features.** Take the MMP data and DNA-binding protein features on Chr V:6120909~6202632 (*ttn-1*) as an example. A) Line plots showing the prediction made from the Flanking sequence preferences of EMS mutagenesis ($P_0$). B) The mutations observed in the MMP dataset. C) Raw ChIP-chip signal data of UNC-62 binding in young adult worms. D) Raw ChIP-chip signal data of UNC-39 binding in worm embryo. E) Raw ChIP-seq signal data of histone H3 in L3 larval worms. F) Raw ChIP-seq signal data of histone H4 in L3 larval worms. G) Line plots showing the prediction made by a Random Forest regressor trained by DNA-binding protein data.
(TIF)

**S3 Fig. Feature importance analysis of the Random Forest regressor.** A-F) Permutation importance (see Materials and Methods) of each feature used to model the Random Forest regressor. The importance is shown in the heatmap alongside the chromosome. Mutation rate is normalized to 0~1 and is shown in the bar plot attaching to the right side of the heatmap.
(TIF)

**S1 Table. Features for Random Forest training.**
(DOCX)

**S2 Table. *C. elegans* Strains in this study.**
(DOCX)

**S3 Table. PCR products for *C. elegans* transgenesis.**
(DOCX)

## Author Contributions

**Conceptualization:** Zhengyang Guo, Guangshuo Ou.

**Data curation:** Zhengyang Guo.

**Formal analysis:** Zhengyang Guo, Yang Wang.

**Funding acquisition:** Guangshuo Ou.

**Investigation:** Shimin Wang, Zi Wang.

**Methodology:** Zhengyang Guo.

**Supervision:** Zhengyang Guo, Guangshuo Ou.

**Validation:** Zhengyang Guo.

**Visualization:** Zhengyang Guo, Yang Wang.

**Writing – original draft:** Zhengyang Guo, Guangshuo Ou.

**Writing – review & editing:** Zhengyang Guo, Guangshuo Ou.

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
