## [Decision Letter · Decision Letter 0]

11 Jul 2024

Dear Dr. Ou,

Thank you very much for submitting your Research Article entitled 'A Machine Learning Enhanced EMS Mutagenesis Probability Map for Efficient Identification of Causal Mutations in Caenorhabditis elegans' to PLOS Genetics. My apologies for the delay. One reviewer was unable to provide comments, and we had to find a new reviewer. 

The manuscript was fully evaluated at the editorial level and by independent peer reviewers. The reviewers appreciated the attention to an important topic but identified some concerns that we ask you address in a revised manuscript.

We therefore ask you to modify the manuscript according to the review recommendations. Your revisions should address the specific points made by each reviewer. In particular, please revise the text related to Machine Learning in such a way that it can be better appreciated by the general audience. 

To resubmit, log into your Editorial Manager account and select the option 'Revise Submission' in the 'Submissions Needing Revision' folder.

Yours sincerely,

Shawn Xu

Guest Editor

PLOS Genetics

Xiaofeng Zhu

Section Editor

PLOS Genetics

Reviewer's Responses to Questions

**Comments to the Authors:**

Reviewer #1: Chemical-mutagenesis forward-genetic screening is a powerful tool in the realm of genetic screening. Forward genetic techniques offer advantages over reverse genetic techniques (RNAi and CRISPR/Cas9) in that it can reveal novel genes and lead to the discovery of unexpected connections and mechanisms. However, identifying the responsible mutation for a given phenotype is laborious and can require a significant time investment.

The authors of this study outline the development of a machine learning tool that can be used to identify causative mutations generated through EMS chemical mutagenesis. They show that the distribution of EMS-induced mutations is non-uniform throughout the genome, both within and across chromosomes; specific nucleotide sequences and different chromatin statuses are biased towards different mutation frequency. This is a key observation that is used to train the Random Forest regressor (Rfr) machine learning tool which can predict what genes give rise to an observed phenotype following mutagenesis. The Rfr model is used to identify the suppressor gene K04F10.2 after EMS-mutagenesis, which can restore ciliary function in OSM-3 mutant C. elegans. Taken together, the conclusions reached in the study are very promising. However, I have a number of concerns about the figures and the explanation of key takeaways as outlined below.

Major points

1) Introduction. The discussion of genetic screening techniques is adequate. Given that the study intersects with the field of machine learning, there should also be a priming outline of machine learning practices that already exist in the field. Add more background for strictly biological researchers who may not be familiar with machine learning techniques and probability mapping.

2) Line 201. Formulation of P0 should also be inserted into the main text since many future panels rely on an understanding of this calculation.

3) Figure 3E/F. It is stated that a distinctive histone binding pattern is not seen in the ‘silent region’ of figure 3 as is the case in the ttn-1 gene. Yet there does seem to be a slight increase in histone binding activity in this region compared to the entire chromosome sequence depicted, particularly in H3. The text does not elaborate on panels E/F specifically, and the conclusion that this pattern is “not observed” is not entirely convincing.

4) Figure 3E/F. There are conflicting results for how histone modification affects mutation frequency in the ‘silent region’ versus the ttn-1 gene. Is it known how “alternative chromatin states” (line 226) affect mutagenesis? Are there experiments that can elucidate the binding patterns at play here? Figure 3 as a whole requires a more detailed explanation with greater emphasis on the rhetorical flow of logic.

5) Figure 5E/F, lines 436-440. The results given in these panels are unclear and appear incongruous with the written text. Did OSM-3 E251K; K04F10.2 double mutants receive the Scarlet-tagged protein or wild type K04F10.2? Is the succeeding condition with Scarlet protein meant to show a diminished capacity of K04F10.2-tagged Scarlet to rescue OSM-3?

Minor points

1) Line 456. typo in “laborintensive”

2) Lines 229-241. Nearly identical paragraphs repeated back-to-back.

3) Line 257. Missing reference for WormBase.

4) Figure 5 legend. Font inconsistency.

5) Figure S1. Misplaced label in panel B.

Reviewer #2: The manuscript by Guo et al presents an innovative and potentially game-changing strategy for chemically-induced forward genetic screening. In the past, mutagenesis-based genetic screens were standard in model organism fields. With technological advances including bioinformatics and genomic engineering, reverse genetic strategies have become the standard to ascertain gene function. Due to the difficulty in identifying causal lesions in mutagenesis screens and ease of knock-down/knockout strategies, forward screens are falling out of favor. However, as authors point out, the power of forward genetic screens is their unbiased nature, their ability to identify novel mutations that may be otherwise missed, and their power to reveal unknown biology. Here, authors use a strategy that capitalizes on the numerous community generated resources (WormBase, MillionMutations project, CHIP-seq, modENCODE) to identify causal mutations in WGS in two proof-of-principle EMS-based screens. In addition to this, authors show that mutation frequency is influenced by nearby sequences and chromatin status. This manuscript will appeal to the readership of PLOS Genetics, with some modification.

As written, the reader needs to be expert in C. elegans, in statistics/machine learning, and in cilia biology. This reviewer is versed in two, but not machine learning and struggled with this section of the manuscript. For example, it would help to more thoroughly explain how the Random Forest Regressor works and why this model was chosen. The same would be true for the non-worm or non-cilia reader in those section. Authors must make the manuscript accessible to the broad readership of PLOS Genetics, so that the impact and usefulness of this powerful strategy can be appreciated and employed by model organism geneticists.

Second, while the data is convincing that this strategy works, how can this be applied in a practical sense? It would be great if authors could work with WormBase/Alliance of Genome Resources to develop a user-friendly interface. I realize this is beyond the scope of this manuscript, but hope this is a future direction.

A few minor things:

Lines 195, 197: typo isare

Lines 230-242 are redundant/garbled

Throughout manuscript: check C. elegans nomenclature for italicizing gene names

line 369 define “back door” (example of needing to be a cilia/kinesin aficionado)

suggestion – request K04F10.2 be named jbts-26

line 434 – Fig. 5E

lines 465, 468: I think kinesin-II is older nomenclature. Heterotrimeric kinesin-2?

Maureen Barr

Reviewer #3: The comments to the authors are provided as an attachment.

**Have all data underlying the figures and results presented in the manuscript been provided?**

Reviewer #1: Yes

Reviewer #2: None

Reviewer #3: Yes

PLOS authors have the option to publish the peer review history of their article (what does this mean?). If published, this will include your full peer review and any attached files.

Reviewer #1: **Yes: **Rui Xiao

Reviewer #2: No

Reviewer #3: No

---

## [Editor Report · Decision Letter 1]

27 Jul 2024

Dear Dr Oh,

We are pleased to inform you that your manuscript entitled "A Machine Learning Enhanced EMS Mutagenesis Probability Map for Efficient Identification of Causal Mutations in Caenorhabditis elegans" has been editorially accepted for publication in PLOS Genetics. Congratulations!

Yours sincerely,

Shawn Xu

Guest Editor

PLOS Genetics

Xiaofeng Zhu

Section Editor

PLOS Genetics

Comments from the reviewers (if applicable):

**Data Deposition**

http://datadryad.org/submit?journalID=pgenetics&manu=PGENETICS-D-24-00407R1

**Press Queries**

---

## [Editor Report · Acceptance letter]

20 Aug 2024

PGENETICS-D-24-00407R1 

A Machine Learning Enhanced EMS Mutagenesis Probability Map for Efficient Identification of Causal Mutations in Caenorhabditis elegans 

Dear Dr Ou, 

We are pleased to inform you that your manuscript entitled "A Machine Learning Enhanced EMS Mutagenesis Probability Map for Efficient Identification of Causal Mutations in Caenorhabditis elegans " has been formally accepted for publication in PLOS Genetics! Your manuscript is now with our production department and you will be notified of the publication date in due course.

With kind regards,

Judit Kozma

PLOS Genetics

On behalf of:
